

# Effects of an 8-week yoga program on sustained attention and discrimination function in children with attention deficit hyperactivity disorder

Chien-Chih Chou and  Chung-Ju Huang

Graduate Institute of Sport Pedagogy, University of Taipei, Taipei, Taiwan

## ABSTRACT

This study investigated whether a yoga exercise intervention influenced the sustained attention and discrimination function in children with ADHD. Forty-nine participants (mean age = 10.50 years) were assigned to either a yoga exercise or a control group. Participants were given the Visual Pursuit Test and Determination Test prior to and after an eight-week exercise intervention (twice per week, 40 min per session) or a control intervention. Significant improvements in accuracy rate and reaction time of the two tests were observed over time in the exercise group compared with the control group. These findings suggest that alternative therapies such as yoga exercises can be complementary to behavioral interventions for children with attention and inhibition problems. Schools and parents of children with ADHD should consider alternatives for maximizing the opportunities that children with ADHD can engage in structured yoga  exercises.

## INTRODUCTION

Attention deficit hyperactivity disorder (ADHD) is a neurobiological condition that commonly occurs among school-aged children, with approximately 5–8% of children being affected. Moreover, the symptoms of the disorder persist into adulthood in up to 60% of childhood cases, meaning that roughly 4% of adults suffer from ADHD (*Barbaresi et al., 2002*; *Froehlich et al., 2007*). The disorder is characterized by inappropriate attention, impulsivity and/or hyperactivity which may cause a variety of problems including academic difficulties, impaired social skills, and strained parent–child relationships (*Harpin, 2005*). In regard to underlying mechanisms of ADHD, several models have been proposed such as maturational lag, cortical hypo-arousal, and developmental deviation (*Barry, Clarke & Johnstone, 2003*). Given that long-term medications can ameliorate ADHD symptoms but may have negative side-effects such as sleep disturbances, reduced appetite, and mood disorders (*Pliszka, 2007*), physical activity can serve as a low-risk treatment for ADHD symptoms. A recent systematic review has suggested that physical activity shows promise as an effective treatment, reporting improvement in measures of interference control, set shifting, consistency in response speed, vigilance and impulsive control among individuals

Corresponding author
Chung-Ju Huang,
crhwang_tpec@yahoo.com.tw

with ADHD (*Halperin, Berwid & O'Neill, 2014*). Therefore, further examining beneficial effects of different forms of physical activity on attention performance in children with ADHD can extend the current understanding of the association between physical activity and cognition function of individuals with ADHD.

Yoga exercise has been found to be a feasible school intervention for children with emotional and behavioural disorders and can be effective in ameliorating the symptoms that also pervasively occur in children with ADHD, such as inattention and bad adaptive skills in class (*Steiner et al., 2013*). Different from normal physical exercise, yoga practicing steers individuals to master certain breathing techniques, postures, and cognitive control which can help promote self-control, attention, body awareness, and stress management (*Kimbrough, Balkin & Rancich, 2007*). Previous studies have demonstrated that yoga shows promise as an intervention for a variety of social, emotional, behavioral, and cognitive ailments (*Diamond & Lee, 2011*). Although previous research regarding the effects of yoga on ADHD symptoms is somewhat limited, several studies have been conducted. Some studies employing yoga as a treatment for ADHD reported beneficial effects such as reduced hyperactivity, inattention, and anxiety, and improved peer relationship quality and sleep patterns (*Harrison, Manocha & Rubia, 2004*; *Jensen & Kenny, 2004*). However, no control group design and a small sample size brought to weaken the validity of these positive effects of yoga intervention on ADHD children's behavior and cognition. By utilizing a larger sample with a symptom-match compared group, the benefits of yoga exercise on alleviating the cognitive deficits of ADHD can be further identified.

Physical activity intervention has been found to derive positive changes in behavioral structures and cognitive function among children with ADHD, which are reflected in reduced impulsivity, anxiety, and improved attention (*Chang et al., 2014*; *Huang et al., 2014*; *Smith et al., 2013*; *Verret et al., 2012*). Additionally, for children with ADHD, even a single bout of exercise has been found to contribute to response preparation (*Chuang et al., 2015*) and task switching (*Hung et al., 2016*), and physical fitness has been associated with baseline cortical activity (*Huang et al., 2015*) and inhibitory ability (*Tsai et al., in press*). The mechanisms of physical activity effects on ADHD children's cognitive function may be due to brain structure changes, enhanced neurotransmitters, and arousal regulation (*Lustig et al., 2009*; *Tang et al., 2008*). Like physical activity, yoga has been found to have beneficial impacts on neurological and physiological activity and behavior in a range of populations. The reported benefits of yoga include increased slow-frequency brain wave activity (*Arambula et al., 2001*); favorable profiles on heart rate (HR) variability, depression, perceived stress, and superior aerobic fitness (*Satin, Linden & Millman, 2014*); and significant decrement of cortisol and increment in brain-derived neurotropic factor (BDNF), serotonin, and dopamine (*Pal et al., 2014*). In essence, the practice of yoga exercise elicits reduced activation of the sympathetic nervous system and increased activation of the parasympathetic nervous system resulting in a sense of equilibrium into the body and mind, and increased emotional self-regulation (*Streeter et al., 2012*). Given that abnormal attention and over-impulsivity characteristics have been considered as major symptoms of ADHD, these previous studies provide compelling empirical evidence for using yoga exercise in ADHD treatment.

Although a few studies have reported potential associations between yoga exercise and improved cognitive functions of children with ADHD (*Jensen & Kenny, 2004*), these findings are required to be replicated to warrant the use of yoga as an effective complementary treatment for this population. Based on evidence revealing deviant patterns in behavioral impulsivity, memory retrieval, sustain attention, stimuli differentiation, and decision making are relevant to ADHD (*Barkley, 1997*; *Bellgrove, Hawi & Robertson, 2006*), the current study further examined the effects of yoga exercise on the sustained attention and discrimination functions among ADHD children. Any changes in sustained attention and discrimination function observed after yoga exercise could be used to identify the effects of yoga on improved ADHD symptoms. Given the results of the studies briefly reviewed above, physical activity including yoga exercise could contribute to reduced hyperactivity and inattention (*Harrison, Manocha & Rubia, 2004*; *Jensen & Kenny, 2004*; *Steiner et al., 2013*), as well as increased interference control, attention shifting, and consistency in response speed among individuals with ADHD (*Halperin, Berwid & O'Neill, 2014*). Reduced hyperactivity, increased attention and response consistency could contribute to better sustained attention while increased interference control and attention shifting should enhance discrimination function which involves inhibition and selective attention.

This study hypothesized that yoga exercise could benefit the sustained attention and discrimination function of children with ADHD by using the Visual Pursuit Test and the Determination Test. The Visual Pursuit Test is usually used to assess visual perception involving sustained attention and the Determination Test is used to evaluate the ability to determine multiple-choice reaction requiring inhibitory ability and selective attention. Although the two tests have not been utilized among individuals with ADHD, their availability has been reported in clinical populations (*Kober et al., 2013*) and young athletes (*Dogan, 2009*). Such a study could be critically important in laying the ground work for both scientific research and clinical application.

## METHODS

### Participants

Fifty participants were recruited via flyers posted in relevant locations, referrals given to the children's parents by their elementary schools, and a number of orientations conducted to introduce the project. All participants were from schools located in suburban areas of a city where most families were the middle to high socioeconomic populations. The inclusion criteria were as follows: children aged between eight and 12 years old who had been diagnosed with ADHD by their own psychiatric physicians and had been confirmed by school pediatricians. All the various subtypes of ADHD (inattentive, hyperactivity/impulsivity, combined) were included regardless of whether he or she was receiving medication for ADHD symptoms. The exclusion criteria were as follows: (a) comorbid conditions such as conduct/oppositional defiant disorder, autism spectrum disorders, or serious affective disorders; (b) a personal history of brain injury or neurological disorders; and (c) currently taking sedatives or other mood altering medications other than the stimulants typically prescribed for ADHD. All the children were assigned to one of two groups according to their school districts: the yoga exercise group ($n = 25$) and the

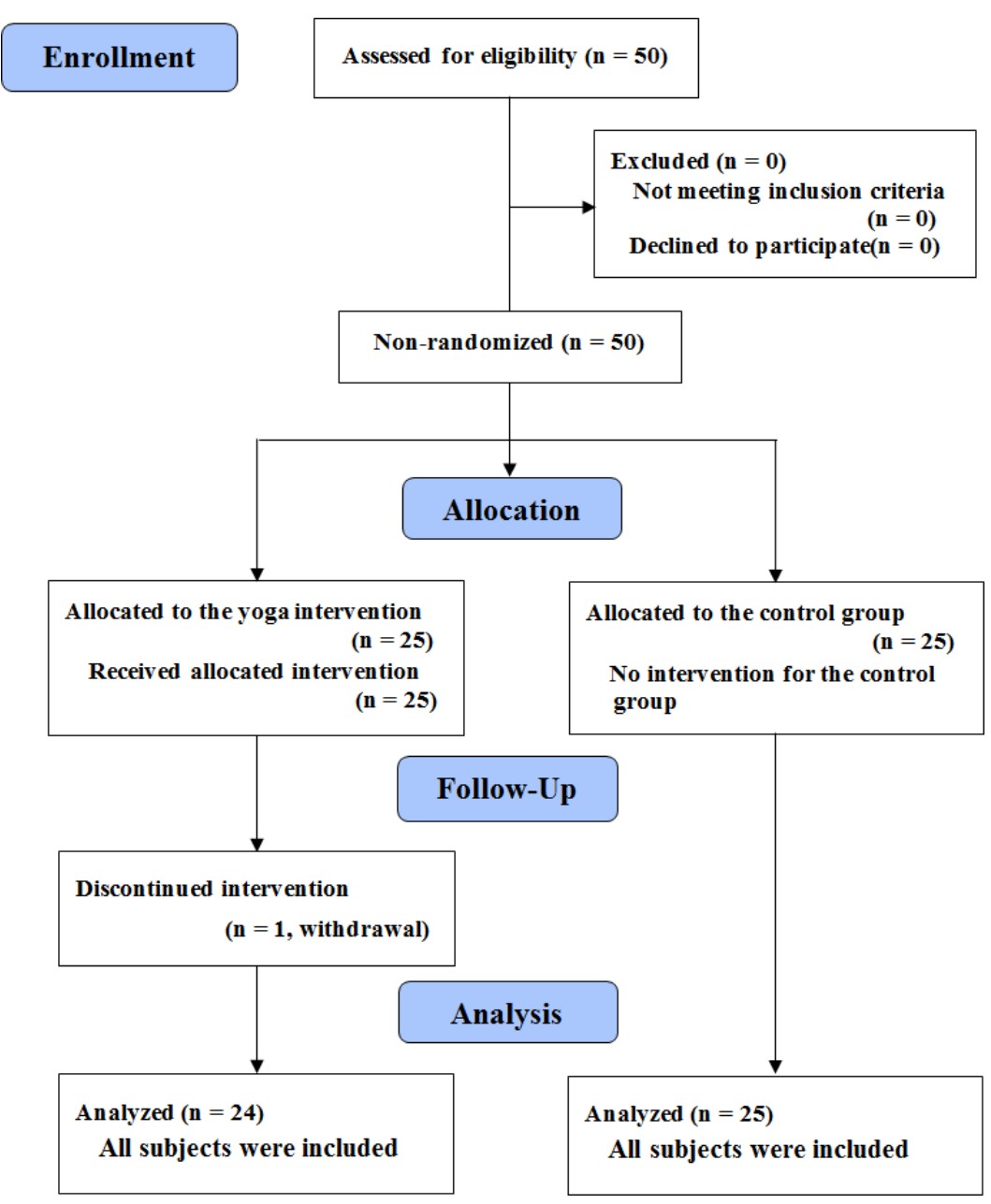

**Figure 1  The flow diagram for the progress of all participants through the trial.**

control group ($n = 25$). One participant withdrew from the yoga exercise group due to personal consideration. Figure 1 shows the flow diagram for the progress of all participants through the study. Participants were instructed to refrain from medications and caffeine intake for at least 24 h prior to undergoing the various tests administered as part of the study. Written informed consent was provided by the parents and the children following a full explanation of the study, which was reviewed and approved by the Research Ethics Committee of National Taiwan University prior to the experiment.
## Measures

### Visual pursuit test

The Visual Pursuit Test of the Vienna Test System (Schuhfried GmbH, Austria), a computerized psychological assessment tool well established in psychological diagnostics (*Schmid et al., 2005*), was used in this study. It is designed as a line tracking test and used for the registration of concentrated targeted perception and selective attention in the visual area. Hence, the performance in this test requires the ability of selective and sustained attention. It consisted of 54 different items; in each item, an array of nine entwined dark lines leading to nine different endpoints was presented on a light background computer screen. The starting point of one out of the nine lines was marked and the participant was asked to follow this line with their eyes to find the corresponding endpoint as quickly as possible by pressing one of nine number buttons on a response panel. The screen was shown for four seconds and then disappeared. Therefore, items to which the participant did not answer correctly or did not respond within the four seconds were reported as incorrect responses. The performance of the participant was scored automatically, considering the number of correct answers and mean RT for correct answers. The test duration for each participant was approximately 10 min per run.

### Determination test

The Determination Test has been used to assess the discrimination ability for reaction speed, attention deficits, and reactive stress tolerance in the presence of continuous but rapidly changing acoustic and optical stimuli (*Shmygalev et al., 2011*). In this study, the participant's task was to react as quickly as possible to visual or acoustic stimuli by pressing the corresponding buttons on the response panel. There were five visual stimuli colored white, yellow, red, green and blue, which appeared in an upper and a lower row on the screen. The reaction buttons assigned to these five colors were arranged on the response panel in such a way that the participant could use both hands. There were two additional visual stimuli, in the form of white, rectangular, visually distinct fields that appeared in the bottom left- and right-hand corners of the screen, to which the participant had to react by pressing the corresponding (left or right) foot pedal. Two acoustic stimuli (high and low tone) were assigned to the two ''sound'' buttons in the middle of the panel. The lower, rectangular black button was pressed for a low tone and the upper rectangular grey button for a high tone. The visual stimuli were presented on the screen and the acoustic stimuli were presented via headphones. The duration of stimulus presentation depended on the respondent's mean RT for the previous eight trials. If the response to a stimulus was not correct, the RT was doubled for the purpose of calculating the duration of the next stimulus. This test contained 180 trials with 20 trials for each stimulus. The number of correct trials was utilized to calculate the accuracy rate and RT of each correct response was reported. The duration for the test was approximately 10 min for each participant.

### Physical fitness

The participants were instructed to not engage in any intense physical activity or take any stimulant medication on the day before the evaluations. In addition to weight and height, the physical fitness of each participant was estimated, including flexibility, muscular

endurance, power, and cardiovascular fitness. The fitness assessment includes measures of flexibility (sit and reach test), muscular endurance (sit-ups in 1 min), power (standing long jump), and cardiovascular fitness (a half-mile run in the fastest possible time). The four subsets of fitness assessment of all the participants were converted into standardized T-scores, and the physical fitness score was computed as the mean of the scores on these fitness subsets.

## Yoga exercise intervention

The manipulation of yoga activity followed the American Physical Therapy Association guidelines for working with children (*Galantino, Galbavy & Quinn, 2008*). Each lesson for yoga activity lasted for 40 min, twice a week for eight weeks, was led by a nationally certified yoga instructor, and was conducted in a dance studio with an average temperature of 24–26 °C. The yoga activity session consisted of a 10-minute stretching and warming-up period followed by a 20-minute yoga activity, which included concentration and balance, improved attention, and breath and body awareness. Finally, each session ended with a 10-minute cooling-down period including balancing, flexibility, and relaxation exercises. This clinical trial was approved by the Chinese Ethics Committee of Registering Clinical Trials. During the entire session, each participant's HR was recorded at one-min intervals by using a Polar HR monitor (Mode ZW 60 GT5; Cardiosport, Waterlooville, United Kingdom). The HR values during the periods of warming-up, main activity, and cooling-down were calculated respectively. The intensity of the main activity was set at 50–60% of maximal HR (HRmax) according to the previous study suggesting that this intensity level of aerobic exercise is beneficial to baseline cognition function of children with ADHD (*Huang et al., 2014*). The HRmax was estimated using a formula "220–age" and then the target HR for each participant was calculated. To monitor the intensity of the main activity, the researchers examined the HR data of each participant during the period of main activity after the completion of each lesson. For all the lessons, the participants reached at least 50% and did not exceed 60% of their HRmax during the main activity. The instructor was recommended to maintain her prescribed activity and intensity.

To test for exercise intensity manipulation, one-way analysis of variance (ANOVA) was conducted to compare the HR differences among the periods of warming-up, main activity, and cooling-down. The results showed a significant increase in HR during the period of main activity for yoga exercise ($105.21 \pm 4.24$ bpm), compared to the periods of warming-up and cooling-down ($78.38 \pm 3.23$, $84.87 \pm 3.18$ bpm), $F_{(2,46)} = 306.04, p < .001$, and partial $\eta^2 = 0.93$. During the main activity for the yoga exercise, the intensity of exercise was approximately 53% of HRmax. Briefly stated, the exercise intensity of yoga activity reached a moderate level.

## Procedure

The participants were invited to come to the laboratory with their parents on two separate days. The children who were undergoing medical treatment were asked to refrain from medication for at least 24 h prior to the experiment. On the first visit, the participant's parent(s) and the participant signed an informed consent form, provided a health history,

and filled out a demographics questionnaire. Each eligible participant then entered the pre-test stage, which consisted of performing the Visual Pursuit Test and the Determination Test and a physical fitness assessment. Prior to the physical fitness tests, the participants and their parents were verbally inquired to assess their or their children's physical readiness for the tests. For those participants who expressed they were not physically well-prepared to take the physical fitness tests, we rearranged the tests for them on other visits as soon as possible. The order of the Visual Pursuit Test and the Determination Test was counterbalanced. For the Visual Pursuit Test, the given participant was asked to find the end of a specified line as rapidly as possible by pressing the corresponding number buttons on the response panel. Before the formal test, eight trials were provided for practicing. If over three trials for which the responses were in error, the participant was asked to repeat the practicing trials until they were familiar with the test. Then, the formal trials were administered.

For the Determination Test, the participant was instructed to perform according to the various color stimuli and acoustic signals presented by pressing the corresponding buttons on the response panel. At the instruction phase, step-by-step instructions gave the participants the necessary information regarding the test. The instructions started by explaining the colored buttons on the response panel. The participants were then introduced to the visual stimuli and sounds; samples of these could be seen and heard by pressing the corresponding buttons. The instruction phase was followed by a practice phase. If more than three errors were made or if no response was made within 45 s on three successive occasions, the practice phase was automatically interrupted and the respondent was instructed to consult the test administrator. The administrator could, if necessary, restart the instruction phase in order to ensure that the instructions were fully understood. Then, the formal trials were conducted to the participants. As these two tests were completed, the height and weight of the participants were measured and the fitness assessment followed after warm-up exercises and detailed explanations of each testing protocol. Participants were allowed to rest between each testing to avoid fatigue.

Participants in the yoga exercise group underwent an eight-week yoga exercise program that consisted of two 40-min sessions per week as an after school program. In contrast to the exercise group, the participants in the control group were simply instructed to maintain their normal life without participating in regular physical activity programs. Within one week of completing the yoga exercise program, all the participants were invited to visit the laboratory for the second time. Each participant was asked to perform the Visual Pursuit Test and the Determination Test again for the comparison of pre- and post-test scores.

## Statistical analysis

To ensure that any potential confounds would be homogenous for both the exercise and control groups, independent $t$-tests or chi-square tests were used to analyze for continuous or discrete scales of demographic data, respectively, to compare between the two groups. Next, the effects of yoga exercise on the performance in the Visual Pursuit Test and the Determination Test were examined by 2 (Group: exercise, control) × 2 (Time: pre-test, post-test) mixed design ANOVAs. Following the ANOVAs, multiple comparisons with Bonferroni–Holm adjustments were applied to control for experiment-associated inflation

**Table 1  Demographic characteristics of the participants.**

| Variable | Exercise group (n = 24; M [SD]) | Control group (n = 25; M [SD]) | Total (n = 49; M [SD]) |
|---|---|---|---|
| Gender (male: female) | 19:5 | 19:6 | 38:11 |
| Age (years) | 10.71 [1.00] | 10.30 [1.07] | 10.50 [1.05] |
| Height (cm) | 143.71 [9.42] | 143.08 [9.43] | 143.39 [9.33] |
| Weight (kg) | 39.88 [7.48] | 40.00 [6.96] | 39.94 [7.14] |
| BMI (kg/m$^2$) | 19.15 [1.97] | 19.50 [2.81] | 19.33 [2.41] |
| IQ (score) | 100.79 [6.96] | 102.32 [9.98] | 101.57 [8.58] |
| Physical fitness ($T$ score) | 49.70 [.50] | 50.48 [3.67] | 50.10 [3.57] |
|    Cardiovascular fitness (second) | 282.90 [28.09] | 289.70 [30.64] | 286.37 [29.31] |
|    Muscular strength (cm) | 128.79 [8.58] | 126.96 [9.91] | 128.35 [9.29] |
|    Muscular endurance (times) | 22.79 [3.41] | 23.12 [3.44] | 22.96 [3.39] |
|    Flexibility (cm) | 25.45 [3.87] | 25.76 [3.89] | 25.61 [3.84] |
| Grade ($n$) | | | |
|    Third | 1 | 4 | 5 |
|    Fourth | 11 | 10 | 21 |
|    Fifth | 7 | 7 | 14 |
|    Sixth | 5 | 4 | 9 |
| ADHD type ($n$) | | | |
|    ADHD-I | 12 | 8 | 20 |
|    ADHD-HI | 7 | 13 | 20 |
|    ADHD-C | 5 | 4 | 9 |
| Medication ($n$) | 10 | 12 | 22 |

Notes.

BMI, body mass index. The IQ (intelligence quotient) score was assessed by the Wechsler Intelligence Scale for Children-IV; ADHD-I, predominantly inattentive subtype; ADHD-HI, predominantly hyperactive-impulsive subtype; ADHD-C, combined hyperactive-impulsive and inattentive subtype.

of type 1 error for small sample sizes. Effect size (ES) values were calculated according to Cohen's d and partial eta-square ($\eta^2$) for the significant main effects and interactions. For all statistical analyses, a significance level of .05 was used prior to the adjustment.

# RESULTS

## Demographic analyses

The analyses results indicated that there were no significant differences between the groups in terms of weight, height, body mass index (BMI), age, intelligence quotient (IQ), cardiovascular fitness, muscular strength, muscular endurance, flexibility, and total physical fitness, $t_{47} < 1.07$, $p > .05$. In addition, in terms of gender, grade, type, and medicine intake for both groups, $x^2 < 2.69$, $p > .05$, suggesting that the two groups were homogenous. Table 1 summarizes the demographic characteristics and physical fitness of the participants in both groups.

**Table 2   Means (SD) for the visual pursuit test and the determination test.**

| Variable | Exercise group | | Control group | |
| --- | --- | --- | --- | --- |
| | Pre-test | Post-test | Pre-test | Post-test |
| Visual pursuit test | | | | |
| Accuracy rate (%) | 95.60 (5.43) | 98.61 (2.95) | 96.45 (3.89) | 95.12 (5.63) |
| Reaction time (sec.) | 1.31 (0.11) | 1.16 (0.12) | 1.34 (0.11) | 1.32 (0.15) |
| Determination test | | | | |
| Accuracy rate (%) | 72.26 (10.63) | 85.48 (11.43) | 75.72 (15.18) | 71.90 (13.80) |
| Reaction time (sec.) | 0.95 (0.07) | 0.86 (0.07) | 0.99 (0.17) | 0.99 (0.13) |

## The visual pursuit test

As the accuracy rate of the Visual Pursuit Test presented in Table 2, a mixed design ANOVA revealed no main effects of Group and Time. However, a significant interaction of Group by Time was found, $F_{(1,47)} = 4.26$, $p = .045$, partial $\eta^2 = 0.08$, exhibiting that the yoga exercise group yielded a higher accuracy rate at the post-test than the control group, $t_{47} = 2.70$, $p = .010$, $d = 0.78$, along with no group differences observed at the pre-test. Furthermore, the exercise group reported an increased accuracy rate after the yoga intervention, $t_{23} = -2.12$, $p = 0.045$, $d = -0.69$, while no change in the accuracy rate was found for the control group, $t_{24} = 0.86$, $p = 397$.

For the RT data, the analysis reported a main effect of Group, $F_{(1,47)} = 12.85$, $p = .001$, partial $\eta^2 = 0.22$, revealing faster RT for the exercise group than the control group. Also, a main effect of Time was observed, $F_{(1,47)} = 12.56$, $p = .001$, partial $\eta^2 = 0.21$, indicating faster RT at the post-test than the pre-test. A significant interaction of Group by Time was found, $F_{(1,47)} = 8.20$, $p = .006$, partial $\eta^2 = 0.15$, demonstrating that the yoga exercise group yielded a faster RT at the post-test than the control group, $t_{47} = -4.18$, $p < .001$, $d = -1.20$, along with no group differences observed at the pre-test. Additionally, the exercise group reported a decreased RT after the yoga intervention, $t_{23} = 4.12$, $p < .001$, $d = 1.29$, while no RT change was found for the control group, $t_{24} = 0.54$, $p = .597$.

## The determination test

In terms of the response accuracy of the Determination Test presented in Table 2, ANOVA revealed a main effect of Time, $F_{(1,47)} = 5.32$, $p = .026$, partial $\eta^2 = 0.10$, indicating a higher accuracy rate at the post-test than the pre-test. A significant interaction of Group by Time was reported, $F_{(1,47)} = 17.48$, $p < .05$, partial $\eta^2 = 0.27$, revealing that the yoga exercise group yielded a higher response accuracy at the post-test than the control group, $t_{47} = 3.74$, $p < .001$, $d = 1.09$, along with no group differences observed at the pre-test. Furthermore, the exercise group reported an increased response accuracy after the yoga intervention, $t_{23} = -5.78$, $p < .001$, $d = 1.22$, while no change in the response accuracy was found for the control group, $t_{24} = 1.15$, $p = 0.263$.

For the RT data, the analysis revealed a main effect of Group, $F_{(1,47)} = 9.29$, $p = .004$, partial $\eta^2 = 0.17$, exhibiting faster RT for the exercise group than the control group. Also, a main effect of Time was reported, $F_{(1,47)} = 4.45$, $p = .040$, partial $\eta^2 = 0.09$, indicating faster RT at the post-test than the pre-test. A significant interaction of Group by Time

was observed, $F_{(1,47)} = 4.79, p = .034$, and partial $\eta^2 = 0.09$, demonstrating that the yoga exercise group yielded a faster RT at the post-test than the control group, $t_{47} = -4.26$, $p < .001, d = -1.25$, along with no group differences observed at the pre-test. Furthermore, the exercise group reported a decreased RT after the yoga intervention, $t_{23} = 4.78, p < .001$, $d = 1.26$, while no RT change was found for the control group, $t_{24} = -0.05, p = 964$.

## DISCUSSION

The findings of this study are consistent with previous research (*Cerrillo-Urbina et al., 2015*), revealing beneficial effects of yoga exercise on the core symptoms of children with ADHD, such as sustained attention and discrimination ability. With HR monitoring applied to ensure a moderate level of exercise intensity, the results showed that the yoga exercise program exerted a positive impact on RT and response accuracy at the Visual Pursuit Test and the Determination Test, whereas no such influences were found for the control group. The results were promising because this study reported the homogeneity of demographic characteristics, intelligence quotient, and physical fitness between the exercise and control groups at the baseline, confirming that these variables might not confound our findings. Also, the participants were recruited from nearby residential areas where the average level of socioeconomic status was similar. In particular, the present findings extend previous research by utilizing a larger sample, symptom-match counterparts, and a moderate-intensity yoga exercise that involved aerobic, flexibility, and perceptual-motor exercises.

In support of our hypothesis, the yoga exercise group demonstrated a faster RT and higher response accuracy on the Visual Pursuit Test than the control group. The result that yoga exercise enhanced selective and sustained attention in children with ADHD aligns with the extant literature regarding effects of alternative therapies on ADHD symptoms (*Majorek, Tüchelmann & Heusser, 2004*; *Peck et al., 2005*). These studies have reported the facilitative impacts of yoga and massages on self-control, relaxation, and concentration for children with ADHD. Yoga exercise typically conducts a variety of poses, deep breath, concentration, and mental and physical relaxation which can positively regulate mental states (*Zipkin, 1985*). It also tends to promote self-control, attention and concentration, self-efficacy, body awareness, and stress reduction (*Peck et al., 2005*). Recently, a meta-analytical study has indicated that yoga exercise suggests substantial improvements such as alleviating impulsivity, anxiety, and social problems and a mild improvement in attention and hyperactivity for individuals with ADHD (*Cerrillo-Urbina et al., 2015*). The present findings are also supported by other research regarding physical exercise effects on children with ADHD (*Smith et al., 2013*; *Verret et al., 2012*), reporting that sustained attention improves with long-term physical exercise. Further, *Palmer, Miller & Robinson (2013)* revealed that after engaging in a bout of movement program incorporating various motor skills, preschoolers exhibited better ability to sustain attention compared to after being sedentary. Their finding highlights the proposition that exercise requiring motor control is more likely to enhance sustained attention for subsequent cognitive task because prefrontal brain regions involved in sustained attention are activated after exercise (*Budde et al., 2008*). Yoga exercise always requires participants to perform complex combinations of
motor skills in a smooth and fluent manner, such as controlling body posture and relative space using vestibular sense (*Peck et al., 2005*). Thus, the yoga program in the current study, which involved posture control and motor skills, might facilitate ADHD individuals' subsequent sustained attention performance of the Visual Pursuit Test by activating the prefrontal cortex.

Moreover, the yoga exercise group exhibited a better performance on the Determination Test than the control group, revealing that yoga exercise could be beneficial to children with ADHD in terms of improving their discrimination ability. Given that the Determination Test requires participants to quickly respond to different kinds of acoustic and optical stimuli, better discrimination ability indicates that the participants exhibit greater inhibition toward the interference of the previous stimulus, as well as faster and more accurate selection toward multiple stimuli and the corresponding reaction (*Dogan, 2009*). Previous studies have provided supporting evidence regarding the positive effects of yoga exercise on the inhibitory function in healthy children (*Telles et al., 2013*). *Telles et al. (2013)* assessed the effects of yoga and physical exercise over three months and found that both interventions improved performance in the Stroop task for healthy children. The cognitive mechanisms involved in this task are attentional vitality and flexibility, as well as inhibition of a dominant response. Therefore, similar to effects of physical exercise, yoga exercise has been postulated to influence attention and inhibition, however, through different pathways. Previous studies also support this notion, reporting that changes in bilateral putamen volumes of the dorsal striatum and globus pallidus after a year long physical exercise were associated with inhibition performance (*Chaddock et al., 2012*). On the other hand, the findings of neuroimaging studies indicated that yoga practitioners exhibited increased blood flow to the dorsolateral prefrontal cortex (*Cohen et al., 2009*). Hence, the benefits of yoga practice in discrimination ability seem to be associated with specific changes in particular brain areas.

Another possibility is that yoga exercise improves discrimination ability by improving attention and information processing. Using a neuroelectrical approach, children with ADHD that have greater motor ability exhibited better allocation of attentional resource and superior efficiency of neuroprocessing than their counterparts with lower motor ability (*Hung et al., 2013*). After an eight-week aquatic exercise program, children with ADHD demonstrated an improvement in accuracy for a widely used cognitive task that assesses behavioral inhibition (*Chang et al., 2014*). The mechanisms underlying these benefits may involve complex neuro-chemical changes and modified functioning of brain areas within the limbic circuit. For instance, yoga exercise was found to be associated with decreased cortisol and increased BDNF, serotonin, and dopamine (*Pal et al., 2014*). These biological mechanisms contribute to attention processing and inhibition in terms of regulating arousal levels in fronto-striatocerebellar circuits and enhancing the control of executive function (*Del Campo et al., 2011*). In animal studies, increased BDNF levels in the hippocampus after exercise were found to be related to elevated learning and memory processes (*Vaynman, Ying & Gomez-Pinilla, 2004*). Although further studies are still required to confirm these findings, previous studies suggest that yoga exercise may result in improved cognitive function in ADHD by the alteration of neuro-chemical expressions.

Overall, the results of this study reported that yoga exercise consisting of breathing manipulations, posture control and body balance, and concentration could enhance sustained attention, interference control, and attention shifting among children with ADHD. These findings suggest that physical activity combining various components of motor skills, body control, and concentration practice is beneficial to specific components of executive function of children with ADHD. Some limitations of this study warrant caution and can direct future research. First, although this study controlled the homogeneity of the two groups in terms of gender, age, ADHD type, BMI, physical fitness, medication, and intelligence, the non-randomized controlled trial design and unbalanced gender proportions (i.e., primarily boys) may limit the validity of the findings. Accordingly, the precise causal effects of yoga exercise on cognitive function in ADHD require future researchers to utilize a randomized controlled trial design. Secondly, medication use as well as time spent in video games, internet, and television watching may confound yoga exercise effects on cognitive function of children with ADHD; further examination is required by ruling out the influences of these factors. Thirdly, based on previous studies revealing a linkage between greater aerobic fitness and better executive function in children (*Telles et al., 2013*), the present study shows little evidence provided to interpret the mechanism of yoga effects on cognitive function of children with ADHD without physical fitness assessment after the 8-week yoga program. Fourthly, although the possibility is low, this study could not completely rule out the possibility that the observed effect was a result of less attention paid to the control group. Finally, future research may investigate and expand upon the present findings by reporting the level of physical activity outside the exercise intervention and applying cognitive and behavioral perspectives to the assessment of ADHD symptoms.

## CONCLUSION

In conclusion, our positive findings are in accordance with previous studies demonstrating that yoga exercise can be utilized as an alternative treatment for children with ADHD to reduce attention and inhibition problems. Furthermore, this study is one of only a few studies that have investigated yoga exercise effects on cognitive function in ADHD. To date, stimulant medications appear to be easy interventions to improve functioning in children with ADHD; however, some documented side-effects prevent parents from implementing them. Thus, yoga exercise shows promise as an effective and low-risk treatment for improving long-term cognitive and functional outcomes of ADHD individuals. Given the beneficial effects of yoga exercise participation on certain important ADHD-related cognitive function, the utilization of yoga exercise as an extra-curriculum or a curriculum is recommended to schools and the parents of children with ADHD.

### Funding

The authors received no funding for this work.

## Competing Interests

The authors declare there are no competing interests.

## Author Contributions

- Chien-Chih Chou conceived and designed the experiments, performed the experiments, contributed reagents/materials/analysis tools, wrote the paper, prepared figures and/or tables, reviewed drafts of the paper.
- Chung-Ju Huang analyzed the data, wrote the paper, prepared figures and/or tables, reviewed drafts of the paper.

## Human Ethics

The following information was supplied relating to ethical approvals (i.e., approving body and any reference numbers):

Research Ethics Committee of National Taiwan University (No: 201302HS003).

## Clinical Trial Ethics

The following information was supplied relating to ethical approvals (i.e., approving body and any reference numbers):

Chinese Ethics Committee of Registering Clinical Trials (ChiCTR-OON-16009017).

## Data Availability

The raw data has been supplied as a Supplemental File.

## Clinical Trial Registration

The following information was supplied regarding Clinical Trial registration:

Chinese Clinical Trial Registry.

## Supplemental Information

Supplemental information for this article can be found online at http://dx.doi.org/10.7717/peerj.2883#supplemental-information.

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
