# Peer review of "Effects of an 8-week yoga program on sustained attention and discrimination function in children with attention deficit hyperactivity disorder"

_PeerJ, doi:10.7717/peerj.2883_

## Round 0.1 · original submission · Major Revisions

I now have received two reviewers' comments. Although both reviewers expressed their interest in your study, several aspects of this manuscript should be revised to improve its clarity. Their observations are presented with clarity so I'll not risk confusing matters by belaboring or reiterating their comments. While I might quibble with the occasional point, I note that I regard the reviewers' opinions as substantive and well-informed. I believe that all of the highlighted reservations require contemplation and appropriate attention in revising the document if it is to contribute appropriately to PeerJ and the extant literature. Please revise or refute according to the two reviewers' comments and provide a point by point reply in addition to the revised manuscript.

Tsung-Min Hung, Ph.D.
PeerJ editor
Distinguished professor
Department of Physical Education
National Taiwan Normal University

·

Basic reporting

Using a non-randomized controlled trial, the current study examined whether participation in an 8-week yoga program benefits sustained attention and discrimination function in children with ADHD. Results revealed that exercise group exhibited enhanced performance in cognitive tasks including Visual Pursuit Test and Determination Test, while no such effect was observed in the control group. The results indicate that an alternative therapy (i.e., a moderate exercise intensity yoga program) might be complementary to behavioral interventions for children with ADHD. Although the current study addresses an important topic with regard to the effect of yoga exercise on cognitive health in children with ADHD, several issues need to be addressed before considering this manuscript for publication.


1. Line 31-32: “for maximizing the opportunities that such children can engage in structured yoga exercises”. Does the “such children” here means children with ADHD or normal children?
2. Line 43-45: "%" and “percentage” here and overall in the text: it is more proper to write them consistently given that these two terms are currently used interchangeably.
3. Line 54-56: “In particular, yoga exercise could be of significant benefit in ameliorating the symptoms of ADHD children”. Please clarify that whether this statement is based on the systematic review mentioned above (Halperin, Berwid, & O’neill, 2014) or a speculation provided by the current author. If this is not based on the review described previously, please provide more insights and discussion to strengthen this statement.
4. Line 87-90: Here, the authors indicated the relationship between yoga and SNS/PNS, citing from one book published 34 years ago. It is unclear what the authors were trying to state regardless of the dated citation. The reviewer assumes that the authors intended to link the association between attention and SNS/PNS and considered Yoga exercise as a mediator. If possible, please add more appropriate recent studies for citations.
5. Line 103-105: Please add references for this statement.
6. Line 107-110: Please add references for this statement. What are those studies? What did these studies find? This information is important for the authors to justify their hypotheses and why they conducted this study.
7. Line 127: “were included so long as the participant met the other criteria”. The “so long as” should be “as long as”.
8. Line 202: “the yoga activity induced arousal which was then slightly reduced after the yoga exercise was ceased”. The authors did not refer to arousal issue or concerns anywhere else in this manuscript. The reviewer was wondering why did the authors bring it up here.

Experimental design

1. Line 129: The two groups were assigned based on school districts (i.e., non-randomized controlled trial), therefore a major concern arising, which weakens the validation of experimental design. In particular, the authors did not provide demographic data including socioeconomic status and did not collect puberty data. That is, the authors failed to validate their results to claim that the improvements in exercise group were due to the intervention other than cognitive maturation and social economic status.
2. Line 127: “all the various subtypes of ADHD were included as long as the participant met the other criteria, and regardless of whether he or she was receiving medication for it”. The description for the recruitment of ADHD children is confusing. What criteria did the authors refer to here? The exclusion criteria or the DSM-IV criteria?
3. Line 173-181: Here the authors indicated that physical fitness were measured and scores were computed as the mean of the scores on fitness subsets. However, was there any case that a participant could not perform normally due to injuries or other factors? It was expected that each participant completes a PARQ (Physical Activity Readiness Questionnaire) prior to fitness assessment. More importantly, given that aerobic fitness has been linked to the cognitive function in children, the authors did not provide fitness subtypes including cardiovascular fitness, muscular strength, muscular endurance, and flexibility in the results although a T-score for a combined fitness level has been reported in Table 1. Without the cardiovascular fitness data reported, it is unclear whether the improvement in the exercise group was due to yoga intervention. Last, the authors did not conduct fitness assessment after 8-week yoga program, in which the fitness may also serve as a confounding variable, weakening the cause-and-effect statement.
4. According to current literature, some other factors could confound the effect of yoga on cognition, such as time spent in video games and screen time. There have been several studies examining the effect of video games on cognitive function. If possible, please add them to the participant information. If the variables were not measured in the study, please discuss these factors in the limitation section.
5. In Table 1, please add notes to explain ADHD-I, ADHD-HI, and ADHD-C, given that these variables are of the inclusion criteria for the participants in this study.

Validity of the findings

1. If possible, please indicate whether puberty, social economic status, and cardiovascular fitness were statistically different between groups in baseline.
2. Line 316-321: The authors cited Budde et al., 2008, suggesting that exercise requiring motor control would enhance sustained attention. They further indicated that the yoga program applied in the current study may facilitate sustained attention in ADHD individuals due to the requirements of posture control and motor skills. To support this argument, the authors may add citations or provide more details of how the yoga program links to motor control. The reviewer expects that the authors may have citations or evidence showing this linkage in either the introduction or the discussion section.

Reviewer 2 ·

Basic reporting

No Comments

Experimental design

No comments

Validity of the findings

No comments

Additional comments

The manuscript described an intervention study that applied an 8 week yoga exercise on a group of children with ADHD and found that yoga exercise resulted in improved performance in both the Pursuit Test and the Determination test. Although generally the manuscript was well written and the conclusion seems supported by their data, there are some issues need to be addressed before the manuscript can realize its potential to contribute to the extant knowledge base between physical exercise and cognitive function in population with ADHD.

General comments
Introduction
1. The introduction could present at least briefly the main current models of ADHD and explain what the present experiments may bring to the deep understanding of the pathology.
2. What is the significance and difference why yoga exercise may bring benefits to children? Author should explain more in depth.
3. In your introduction, I can understand that yoga may improve sustained attention, but how it affects discrimination functions should be stated more clearly in introduction.
4. P4 L68-70 The statement of “no control group design and a small sample size brought to question the feasibility of these positive effects of yoga intervention on ADHD children’s behavior and cognition” look odd because the problem of no control and small sample size should be more detrimental to the validity and reliability of the finding rather than the feasibility of these positive effects of yoga intervention on ADHD children’s behavior and cognition.
5. P6L104 & P17L315 “sustain attention” should be “sustained attention” Please provide citation for the cognitive components involved in the Pursuit Test and the Determination test. In addition, it’d be helpful to cite studies that showed that children with ADHD performed more poorly on these two tests.
6. There are some redundancies for the two paragraphs starting from P3L57 and P5L94. Please integrate them.

Method
1. Physical Fitness
You converted physical fitness score into standardized T-scores. For me, I think the score should covert according to age, since you include age between 8-12 children. Did you consider using norm to covert the score?
2. Yoga Exercise Intervention
You used a Polar HR monitor to collect HR during the entire yoga session. But you never mention about how you define or control the intensity of yoga exercise? And why was this intensity used? Furthermore, you used HRmax (220 - age) to represent the intensity, did you consider using HRR (heart rate reserved).
3. What kinds of IQ test have been used in this study?


Results
1. Statistic
In general, when describing the results of ANOVAs, the authors should always begin with the main effects and then describe as briefly as possible the interactions when one of them is significant.

Discussion
1. My main concern here relates to the conclusions made by the authors how to explain your data and finding. The authors try to link “Discrimination ability and inhibition”, which for me are not substantiated by any of their data. Authors should explain more clearly about this test represent which cognitive function, or executive function, specifically.
2. In general, the discussion is very descriptive and does not reframe the data within the big models of executive functions that were mentioned in the introduction section. The authors do not either discuss their data within the frame of the major current theories put forward to explain ADHD. A much broader perspective should be taken for this discussion to be really interesting, beyond the very factual elements that are given in the present version of the manuscript.
3. Again, the authors try to emphasis that you used moderate intensity in yoga exercise. These bring me more questions about how you define your intensity? And is 53% HRmax a moderate intensity?

Conclusion
Given that no comparison between medication and yoga were performed in the present study, it is too early to claim that yoga can be used as a remedy for children with ADHD.

---

## Round 0.2 · accepted · Accept

I have now received two reviewers’ comment and both reviewers were satisfied with your reply and revisions from previous comments. You and your coauthors have my congratulations. Thank you for choosing PeerJ as a venue for publishing your research work and I look forward to receiving more of your work in the future.

Tsung-Min Hung, Ph.D.
PeerJ editor
Distinguished professor
Department of Physical Education
National Taiwan Normal University

·

Basic reporting

No Comments

Experimental design

No Comments

Validity of the findings

No Comments

Additional comments

The authors have completely addressed the questions provided by the reviewer and have completely revised the manuscript. The reviewer considers this manuscript now is in good shape and has no further questions/concerns on this manuscript. Congratulations to the authors for their hard work and the effort they have made.

Reviewer 2 ·

Basic reporting

No Comments

Experimental design

No Comments

Validity of the findings

No Comments

Additional comments

Please check the APA format through article.